# Provider perspectives on screening and treatment for opioid use disorder and mental health in HIV care: A qualitative study

**Brandon A. Knettel**[1,2]*, **Hillary Chen**[3], Elena Wilson[4], David Agor[5], Mehri S. McKellar[6], Susan Reif[2,4]

1 Duke University School of Nursing, Durham, North Carolina, United States of America, 2 Duke Global Health Institute, Duke University, Durham, North Carolina, United States of America, 3 Duke Department of Population Health Sciences, Durham, North Carolina, United States of America, 4 Center for Health Policy & Inequalities Research, Duke University, Durham, North Carolina, United States of America, 5 University of Pennsylvania School of Nursing, Philadelphia, Pennsylvania, United States of America, 6 Duke University School of Medicine, Durham, North Carolina, United States of America

* brandon.knettel@duke.edu

**Data Availability Statement:** A de-identified dataset may be made available upon request from the corresponding author after obtaining an appropriate data transfer agreement. Requests

## Abstract

### Background

HIV, opioid use disorder (OUD), and mental health challenges share multiple syndemic risk factors. Each can be effectively treated with routine outpatient appointments, medication management, and psychosocial support, leading implementers to consider integrated screening and treatment for OUD and mental health in HIV care. Provider perspectives are crucial to understanding barriers and strategies for treatment integration.

### Methods

We conducted in-depth qualitative interviews with 21 HIV treatment providers and social services providers (12 individual interviews and 1 group interview with 9 participants) to understand the current landscape, goals, and priorities for integrated OUD, mental health, and HIV care. Providers were purposively recruited from known clinics in Mecklenburg County, North Carolina, U.S.A. Data were analyzed using applied thematic analysis in the NVivo 12 software program and evaluated for inter-coder agreement.

### Results

Participants viewed substance use and mental health challenges as prominent barriers to engagement in HIV care. However, few organizations have integrated structured screening for substance use and mental health into their standard of care. Even fewer screen for opioid use. Although medication assisted treatment (MAT) is effective for mitigating OUD, providers struggle to connect patients with MAT due to limited referral options, social barriers such as housing and food insecurity, overburdened staff, stigma, and lack of provider training. Providers believed there would be clear benefit to integrating OUD and mental health treatment in HIV care but lacked resources for implementation.

may be directed to the corresponding author. The data may also be made available upon request to the Center for Nursing Research at the Duke University School of Nursing at doche002@mc.duke.edu or 919-668-3836. Sharron L. Docherty, Vice Dean of the Center for Nursing Research, is the primary point of contact for the Center for Nursing Research.

**Funding:** This work was supported by the Duke Center for AIDS Research [P30 AI064518], an NIH funded program, in the form of a grant to BAK.

**Competing interests:** The authors have declared that no competing interests exist.

## Conclusions

Integration of screening and treatment for substance use and mental health in HIV care could mitigate many current barriers to treatment for all three conditions. Efforts are needed to train HIV providers to provide MAT, expand resources, and implement best practices.

## Introduction

In 2021, opioid overdose contributed to 80,816 deaths in the United States, making it the leading cause of injury death in the nation [1, 2]. In addition, opioid misuse can impact well-being, increasing the risk for homelessness, unemployment, mental health challenges, and sexually transmitted infections, including HIV [3–7]. Relationships between HIV, opioid use disorder (OUD), and mental health challenges are complex and syndemic. People living with HIV (PLWH) are more likely to experience chronic pain compared to uninfected people, even when their HIV is well managed, likely due to chronic inflammation and social challenges that are more common among PLWH; one study found that nearly half (48.7%) of PLWH struggle with daily pain [8].

Because of challenges with chronic pain, PLWH are more likely to be treated with opioids, creating increased risk for addiction [9, 10]. Simultaneously, PLWH who use opioids are more likely to experience challenges across the HIV care continuum, including delayed entry into care, suboptimal linkage to treatment, and lower adherence to antiretroviral therapy [11]. HIV and OUD also share multiple social determinants–including low health literacy, poverty, and exposure to violence–that can contribute to poor treatment outcomes such as increased disability, morbidity, and mortality [12, 13].

States in the U.S. South have been disproportionately impacted by both HIV and opioid use disorder [14]. Mecklenburg County, North Carolina (NC) is designated as a priority jurisdiction in the national plan for Ending the HIV Epidemic (EHE) [15], with HIV prevalence nearly three times higher than state and national rates [16]. The county also faces a considerable burden in the opioid epidemic, with an estimated 165 opioid overdose deaths in 2020, which is by far the greatest of any substance measured [17]. Yearly opioid overdose deaths in North Carolina more than quadrupled from 2012 to the end of 2021, including a dramatic spike during the COVID-19 pandemic [17]. Mecklenburg is a large county that includes the greater Charlotte metropolitan area, surrounding suburban areas, and several rural communities, each of which faces unique challenges in improving access to screening and treatment for both HIV and OUD.

The most effective intervention for OUD is medication-assisted treatment (MAT) combining medications for opioid use disorder such as buprenorphine, naltrexone, and methadone, along with psychosocial therapy [18–21]. HIV and OUD are both chronic conditions that are treatable with daily medication, with outcomes enhanced by strong psychosocial support [22, 23]. Further, both HIV and OUD have frequently been linked to mental health challenges. For example, experiencing prior trauma is a risk factor for both HIV and OUD [24]. Other mental health conditions, including depression and anxiety, may be present before HIV infection or the onset of opioid use, often with a worsening or progression, or may develop after an HIV or OUD diagnosis [25, 26].

The common comorbidity of HIV, OUD, and mental health challenges have led implementers to consider integrated treatment for these conditions under one roof, and some efforts have been made to improve access to mental health care in the context of HIV care [27]. The first step toward integration of mental health and OUD treatment in HIV care is effective, universal screening of these conditions at every HIV care appointment [25]. However, several

barriers to screening and integration have been identified, including lack of provider aware-
ness and time and resource constraints [28, 29]. A systematic review identified six studies
examining the integration of treatment for OUD in HIV care settings, five of which focused
on MAT and one of which offered counseling only [27]. These studies demonstrated several
positive clinical benefits for both HIV and OUD, included improved initiation of antiretroviral
therapy (ART), decreased opioid use, decreased needle sharing, and improved health-related
quality of life [27]. Another recent review identified benefits for HIV viral suppression [30].
However, the integration of OUD treatment in HIV care did require considerable investment,
including commitment from leadership to support multidisciplinary care teams, up-to-date
provider training, and sufficient pharmacy stock for substance use treatment [30]. Additional
barriers included added costs for labor, facilities, and urine toxicology testing, as well as a
higher burden of treatment for existing HIV specialists [27]. Although studies have emerged
demonstrating the benefits of integrated OUD treatment in HIV care, few have addressed the
common comorbidity with mental health challenges.

The objectives of this study were to further explore the current landscape of screening and
treatment for OUD and mental health challenges in HIV care, including potential benefits and
barriers related to the implementation of integrated screening and treatment. To achieve these
objectives, we interviewed HIV treatment providers and social service providers who are cur-
rently offering services related to HIV and OUD in Mecklenburg Country, North Carolina.

## Methods

The study team conducted in-depth qualitative interviews with a purposive sample of medical
and social service providers offering treatment for people living with HIV and OUD in Meck-
lenburg County, North Carolina. The team first compiled a list of known local providers
engaged in HIV-related care and/or OUD treatment, and contacted each to invite them to par-
ticipate in a group or individual qualitative interview. Preliminary lists were compiled from
our experience conducting prior research in the area and online searches of HIV and OUD
treatment providers. HIV-related care included HIV testing, antiretroviral medication man-
agement for PLWH, and pre-exposure prophylaxis (PrEP) for people with elevated risk of HIV
infection. Upon completion of their interviews, recruited participants were asked to name
other medical and social services providers in the county and these providers were then con-
tacted and invited to participate. We reached out to 15 medical and social services providers,
all of whom either completed an interview or referred us to other professionals within their
organization who completed an interview.

A member of the research study team trained in qualitative interviewing conducted 30–60
minute virtual in-depth individual interviews (IDIs) with 12 providers using a semi-structured
interview guide on videoconferencing software. Nine additional participants at one large site
participated in a 35 minute virtual group interview. Interviews took place between May 3,
2022 and October 24, 2022. Prior to each interview, participants completed a brief verbal ques-
tionnaire to collect demographic and background information including their professional
experience, current role, and characteristics of their organization. All participants signed an
online consent form prior to participating. No compensation was provided for participation.

The interview guide provided language for initial questions and prompts for additional
probes, as well as flexibility for the interviewer to probe additional lines of inquiry based on
their judgment. In instances where an area of inquiry did not apply to an interviewee, that sec-
tion was skipped. All interviews were audio recorded, transcribed and de-identified, and
uploaded to a secure online database for analysis. The study was determined exempt by the
Institutional Review Board of the Duke University Health System.

### The qualitative interview guide

The development of the interview guide was informed by the Implementation Research Logic Model (IRLM) [31, 32], which considers contextual determinants, including compatibility with existing care models and stakeholder acceptability, strategies to address identified barriers, and mechanisms for outcome improvement. Data from the study are intended to inform future implementation strategies to improve OUD and mental health care in the context of HIV treatment, which may include OUD/stigma education and harm reduction training for HIV clinic personnel, introducing OUD and mental health screening in HIV care, integrating mental health treatment into care, and training HIV clinic providers to offer MAT. A summary of the components, questions, and prompts explored in the semi-structured interview guide, with associated IRLM elements, can be found in Table 1.

### Qualitative analysis

Prior to commencing recruitment, the research team discussed personal reflexivity, including personal attributes and prior knowledge, experience, and assumptions related to HIV, OUD, and mental health. The research team included content experts in each of these areas, and we conducted a one-hour team training on cultural humility, stigma, and harm reduction principles in research. Interviews were analyzed using an applied thematic approach [33] and NVivo 12, a qualitative data software program, to code the data. Thematic data analysis included a combination of deductive and inductive coding, with a priori themes based on the qualitative interview guide utilized for the development of broader categories in the codebook with the emergence of subcodes developed from rereading the collected data transcripts. Two interviews were initially coded by multiple team members, who then came together to discuss and refine the codebook, add emergent theme categories, and resolve any discrepancies of interpretation of the data to ensure validity. The remainder of interviews were then coded using line by line analysis.

The study sample size was determined by the number of eligible providers in Mecklenburg County and considerations of data saturation, the number of new themes emerging in new interviews during preliminary analysis [34].

We randomly selected three individual interviews and the group interview to be re-coded by a second reviewer and evaluated for inter-coder agreement using a pre-established threshold of 80% agreement [35]. Inter-coder agreement on these transcripts was 87.7% (range 85.7–88.9), which exceeded the desired threshold.

## Results

### Interview participants

Interviews were conducted with 21 health care and social service providers currently offering services related to HIV and/or substance use at 11 unique facilities in Mecklenburg Country, NC. We conducted one group interview with 9 participants and 12 individual interviews with 15 social services providers and 6 medical providers, including outreach workers, social workers, case managers, clinical program coordinators, peer specialists, outreach workers, clinic directors, medical doctors, and a physician assistant. The majority of participants (67%) were ages 30–49 years, with more representation (62%) from people identifying as female than male and two-thirds (67%) identifying as Black/African American. Interviewees had been in their current role for an average of 5 years, with a range of 1 to 15 years (Table 2).

**Table 1. Summary of semi-structured interview guide.**

| Domain | Area of Inquiry | IRLM Core Element(s) | Questions and Prompts |
|---|---|---|---|
| Patient Characteristics | Patient population | Implementation Determinants | • Population served<br>• Mental health among patients<br>• Substance use among patients<br>• Opioid use among patients<br>• Patient interest in MAT<br>• Challenges to treatment accyess |
| Current Clinic Services Related to HIV, Substance Use, and Mental Health | HIV testing, treatment and services currently offered | Implementation Determinants | • Treatment and services currently offered at the clinic<br>• Treatment and services offered elsewhere in the community<br>• Referral process and challenges |
| | Screening for substance use | Implementation Strategies and Mechanisms | • Whether screening tool is used<br>• Which patients screened<br>• Who administers screening<br>• Screening follow up<br>• Patient comfort being screened |
| | Substance use treatment and services currently offered | Implementation Determinants | • Treatment and services currently offered at the clinic (if any)<br>• Treatment and services offered elsewhere in the community<br>• Referral process and challenges |
| | Mental health treatment and services currently offered | Implementation Determinants | • Screening, treatment, and services currently offered at the clinic (if any)<br>• Treatment and services offered elsewhere in the community<br>• Referral process and challenges |
| | Strategies and resources needed to improve or expand services | Implementation Strategies and Mechanisms | • Substance use and mental health screening<br>• To improve or expand existing services<br>• To improve referral processes for external services<br>• To offer new services<br>• To support clinic with resources and provider training to offer MAT |
| Provider experiences related to MAT in the context of HIV-related care | | Implementation Mechanisms and Outcomes | • Provider training<br>• Motivation/interest in offering MAT<br>• Challenges to providing MAT<br>• Benefits of providing MAT<br>• Clinic support for offering MAT |

*Note.* MAT: medication-assisted treatment for opioid use disorder

**Table 2. Participant demographics (N = 21).**

| Demographic Characteristics | n (%) |
|---|---|
| **Gender** | |
| Female | 13 (62%) |
| Male | 8 (38%) |
| **Age (in years)** | |
| 18–29 | 1 (5%) |
| 30–49 | 14 (67%) |
| 50–64 | 6 (29%) |
| **Race** | |
| Black or African American | 14 (67%) |
| Non-Hispanic white | 6 (29%) |
| Both white and Black or African American | 1 (5%) |

## Organizational characteristics

Participants provided information on characteristics of the clinic or organization where they work. These included five community nonprofit organizations that provide HIV treatment and prevention services, two Federally Qualified Health Centers (FQHCs), two clinics that provide both primary care and infectious diseases care, one hospital infectious diseases clinic, and the county Health Department. HIV-related care typically included HIV testing, antiretroviral medication management for PLWH, pre-exposure prophylaxis (PrEP) for people with elevated risk of HIV, and management of common comorbid health conditions such as hepatitis. At several clinics, HIV providers also offered primary medical care for conditions unrelated to HIV. All of the clinics work with uninsured patients and almost all accept Medicaid/Medicare.

Only one clinic offered in-house MAT for patients with OUD, in the form of sublingual buprenorphine/naloxone, which was also a clinic that offered both primary care and infectious diseases care. A provider at this clinic noted "increasing amounts of substance use in general and increasing demand for our MAT program," with 30% growth in their MAT patient population in the prior year. As demand has increased, they have hired more providers capable of providing MAT to help meet this need. All of the clinics that did not offer in-house services did offer linkages or referral to outside services such as housing assistance, mental health, substance use treatment, food support, and harm reduction services. No participants reported having peer support workers at their clinic. Roughly half had social workers and/or case managers.

With regard to mental health treatment, three clinics had an in-house provider whereas about half of participants described referring patients with mental health needs to outside clinics or community organizations that can provide needed care. Most participants felt there would be a clear benefit to adding more integrated services within their clinic, including support for substance use mental health treatment, and MAT. The primary barriers to adding these services were lack of financial support and human resources to implement new programs. For example, several participants mentioned actively seeking to hire providers to offer mental health care in-house, but experiencing challenges in finding candidates for these positions. As a result, most clinics currently prioritize providing HIV-related care and treatment specifically.

## Substance use among patients engaged in HIV-related care

Nearly all of the study participants described substance use and mental health challenges as common among patients seeking HIV-related care. Most interviewees highlighted alcohol, marijuana, and sometimes methamphetamine and crack or cocaine as substances they see most frequently among the populations they serve, whereas opioids were emphasized less often. When asked specifically about opioid use among their patients, many participants focused their responses on injection opioid use.

> "*In regards to substance abuse, crack and cocaine have always been a thing and heroin has always been a thing. Marijuana has always been a thing. But as the years have gone by, you saw people mixing different drugs together and they're more addictive in less time. So a lot of times, when you're talking to clients, things that they used to be able to pull away from, they can no longer pull away from.*"

When asked about other routes of administration including misuse of prescription opioid medications, many participants acknowledged that this is likely a much more common, but hidden, problem.

Despite the perception that drug use is very common among their patients, more than half of participants stated that they do not assess for substance use during HIV-related appointments or that assessment is not done consistently. Among those who do assess for substance use, most stated they do not use formal assessments, but ask the questions informally or that they only ask about substance use if it seems relevant to the patient's HIV care. For example, one participant stated, it is a "clinician-to-clinician decision about how they like to screen and talk about [treatment] options." More than half of participants identified improving or standardizing their screening procedures as an area for improvement of services.

Among patients who use drugs, participants expressed that there is commonly a lack of awareness and acceptance among patients of the need for treatment to manage their use. When asked whether specific subsets of their patient population are more likely to be experiencing challenges related to substance use, several participants indicated that sexual and gender minority (SGM [LGBTQIA+]) patients (especially men who have sex with men [MSM] and transgender individuals) and unhoused patients are groups that seem to be disproportionately affected by substance use.

## Other challenges influencing engagement with HIV-related care

Nearly all of the participants described mental health challenges as common among patients in HIV-related care and reported that mental health challenges often co-occur with substance use. Common mental health challenges included depression, posttraumatic stress, and anxiety. Participants also described seeing patients with cognitive impairment and psychosis, both of which can occur due to chronic drug use.

> "*[Our patients] are very chronic, long-term substance users oftentimes, and these are usually individuals that also may have cognitive, as well as mental health diagnoses, which makes it a lot more challenging for them to manage all of that at the same time.*"

Participants described mental health challenges as having a significant negative impact on HIV care engagement, including disruptions to long-term retention in care, consistency in attending appointments, and adherence to antiretroviral medication and other treatments. This created a cyclical pattern of deteriorating physical health and mental health that, especially when coupled with substance use, could lead to likely increases in morbidity and mortality.

Participants identified several other common barriers to HIV patients' engagement with treatment and overall health, often related to finances and social challenges. One participant described it in this way; "When you take mental health and substance use, you've got the main barriers for our patients, followed by housing insecurity, food insecurity, transportation, and then criminal justice involvement."

Participants noted there is a lack of clinic and community resources available to support individuals with social challenges such as housing insecurity. Some clinics provide support to coordinate and link patients to housing, prevent patients from being evicted from their homes, assist with food, or to cover utility costs. Several interviewees explained that without stable housing, patients with HIV, substance use disorders, and other mental health challenges often struggle to stay consistently engaged in their HIV-related care and to seek substance use or mental health treatment.

> "*The resources [for housing] just aren't there in a lot of cases. . . It's kind of like the hierarchy of needs. Without housing, it's difficult for them to seek mental health services or substance*

*use services because their main concern at that point in time is housing. Even their health goes on the back burner because they just want a place to stay."*

Many participants also highlighted transportation as a significant barrier to accessing services. Some noted that their clinics can offer transportation vouchers, cover costs using outside funding, or offer services via telemedicine to ameliorate this challenge. Nevertheless, transportation challenges continue to be a common reason provided by patients for missing appointments or falling out of treatment.

Several participants discussed financial barriers to care, noting that some clinics in the area do not accept patients without insurance, do not provide Medicaid/Medicare coverage, or require payment beyond patients' capabilities even with sliding scales. These financial challenges extend to medication prescription coverage, as well as substance use and mental health care. Interview participants explained that though there are a few social services programs in the area that subsidize copays for mental health care or substance use or provide services free of charge, availability is extremely limited and has inclusion criteria for qualification.

Finally, more than half of the participants described a negative impact of stigma related to mental health and substance use that can prevent patients from discussing substance use or mental health challenges with providers or other support systems, or from asking about services that may be available to them. A few interviewees described stigmatizing comments they have heard or behaviors they have seen staff engage in and noted that this can negatively impact patients' experiences in care. Multiple participants identified cultural humility and stigma reduction as areas in which their organizations could benefit from training.

## Areas to improve substance use and mental health treatment in HIV-related care

More than half of participants discussed challenges related to referring patients for mental health and/or substance use services. Some of these challenges include long waitlists, difficulty following up with patients to ensure they were able to connect to care, lack of available services to refer to (especially for patients with financial barriers), and the burden of building trust with another provider that patients may engage with when being referred out for care. Given these challenges with referrals, some HIV providers attempted to provide counseling and prescribe psychotropic medications, but they often encountered cases they did not have the expertise or capacity to manage in-house.

*"There is a significant number of people with untreated mental health issues, and our providers are not comfortable treating anything more than like minor depression. They're okay putting them on like a low dose SSRI. They're not doing any kind of full psychiatric evaluation and they're definitely not prescribing anti-psychotic drugs."*

Participants acknowledged that the lack of referral options was a complex problem that required large scale, systems-level interventions, such as increased government support to increase the availability of substance use and mental health services and improve access to existing treatment options.

With limited external referral options, several interviewees noted how beneficial it is when a clinic can provide HIV-related care, primary care, mental health, and substance use treatment all under one roof.

*"It would be great if substance use [treatment] was in-house for us, because when we have a client that is interested in treatment that tends to be where we lose them, because they don't*

*want to have to go to this agency and that agency and this place and that place. If they could just come to a central location with us, I think that we would have a bit more success in getting clients to engage."*

Nearly all of the participants discussed the need for more provider training to enhance knowledge on how to treat addiction, raise awareness of services available in the community, increase cultural humility, and reduce stigma when working with patients experiencing substance use and/or mental health challenges.

*"More training. That is definitely needed. Sensitivity training, especially towards substance use. I hear a little undercurrent every now and then where people make comments about drug use, as if it is some deliberate diagnosis that they have a choice in. Everybody needs it."*

Participants felt that providers would be open to more education. However, many interviewees also expressed concern about current staff workloads and the need for more staff to take on new initiatives focused on improving linkage to substance use and mental health treatment.

Several participants explained that having peer navigators to support patients would be beneficial for facilitating trust, helping patients to access needed services, and successfully referring patients for mental health and substance use care.

*"I think that [patients] would be definitely more comfortable talking to people in outreach or somebody who's a peer that can just talk about their experiences. . .I think that people are more likely to open up if it's somebody who they know is in the community as well."*

## Discussion

In these interviews with healthcare and social services professionals providing HIV-related care, there was general agreement about the challenges of comorbid substance use and mental health challenges, the need for improved services in these areas, and the benefit of integrated HIV, substance use, and mental health treatment under one roof. There is an emerging evidence base demonstrating the value of integrated substance use treatment in HIV care [27, 30]; however, efforts at implementing these findings in routine clinical care remain nascent and little attention has been paid to syndemic patterns of HIV, substance use, and other mental health challenges.

The U.S. government's plan for Ending the HIV Epidemic (EHE), introduced in 2019, includes a goal of a 90% reduction of new HIV infections in the U.S. by 2030 [15]. To achieve this goal, the plan includes an emphasis on priority populations and jurisdictions with high HIV prevalence, increases in funding for HIV prevention and treatment, and addressing key barriers to progress in ending the epidemic, "including trends in injection and other drug use; HIV-related stigma; homonegativity and transnegativity; lack of access to HIV prevention, testing, and treatment; and a lack of awareness that HIV remains a significant public health threat" [15]. The site of this research, Mecklenburg County, North Carolina, was identified as one of 48 counties with the highest number of new HIV diagnoses to be prioritized in the EHE plan [15].

In the current research, substance use was described as extremely common among patients seeking HIV-related services when screening did occur, with an increasing burden of OUD in this population. However, few of the participants' clinics conducted standardized evidence-

based screening for substance use with valid psychometric scales, and only one clinic provided in-house substance use treatment and MAT. Providers who did not conduct routine screening rarely asked about opioid use in their informal substance use assessments, reinforcing OUD's status as a hidden epidemic in many healthcare settings [36].

These findings add to a large and growing body of literature highlighting the challenge of OUD in HIV care and failures of the health system to adequately respond by improving access to OUD treatment [11, 37, 38]. MAT is an effective treatment for OUD, involving daily medication to reduce opioid cravings, supplemented by behavioral support, adherence counseling, and education [39, 40]. Prior to 2023, providers required specific training and an associated waiver to prescribe medications for OUD, such as buprenorphine; however, these requirements have now been waived, removing a crucial barrier to providers in all settings offering these safe and effective treatments [41]. It will be crucial in the coming months to capitalize on the improved regulatory environment by seeking to bring evidence-based models for OUD treatment to scale within HIV treatment settings.

Mental health treatment is another crucial aspect of care that can improve quality of life, reduce risk of acquiring HIV or initiating substance use, and improve outcomes of PLWH and people who use opioids [42, 43]. In the current study, participants reported barriers to mental health care that hinder treatment for HIV and substance use, with few clinics offering formal mental health screening and limited in-house options for treatment. These challenges were further exacerbated by poor external referral options for mental health care, including a lack of providers, long wait times, and high costs of treatment. Further, participants expressed concerns about HIV stigma that might occur in the health system and the desire to know and trust providers before sending referrals to them. For PLWH who are also struggling with a substance use disorder and/or another mental health disorder, each condition brings the potential for stigma that can impact social support, disclosure, care engagement, and health outcomes [44, 45].

Participants identified several common social concerns that often co-occurred with HIV, substance use, and mental health challenges, including housing insecurity, food insecurity, and criminal justice involvement. Whether these challenges existed prior to receiving an HIV diagnosis or arise afterward, they must be addressed to increase the likelihood of long-term treatment adherence and positive health outcomes [46–48]. Both HIV care and OUD treatment rely strongly on attention to behavioral health, daily medication adherence, health education, and addressing social determinants to maximize the potential for treatment success [47, 48]. Therefore, it is logical, resource efficient, and imperative to combine treatment approaches for HIV and OUD under one roof [27, 30, 49].

Participants in our study identified integrated treatment for HIV, OUD, and mental health as valuable and desirable, but also difficult to implement due to current providers and staff feeling overburdened and lacking resources to develop new programs. At the national level, some important steps have been made to open the door for improved treatment integration. These include the elimination of the waiver requirement for prescribing medication for OUD [41], new government funding and drug company settlements to address the opioid epidemic [50], and increased support for HIV treatment through the EHE plan [15]. However, sustained efforts are needed to ensure this support reaches providers and patients through evidence-based interventions, and integrated treatments among high-risk groups should be at the top of this list. This will require investments at all levels of the healthcare workforce to increase treatment capacity, including the task-sharing, task-shifting, to peer navigators and community health workers [51]. Telehealth treatment may also be a promising direction to maximize resources [27, 39, 51].

Findings from this study should be interpreted in light of the following limitations. We recruited participants from diverse organizations providing HIV-related services in Mecklenburg County, North Carolina; however, perspectives may not be representative of the broader professional community in the state or elsewhere. Future analyses may wish to assess differences in service provision based on size and type of facility. Researchers may also wish to examine potential differences in themes observed in individual versus group interview formats. As care providers were the subjects of these interviews, information related to HIV and OUD treatment participation for community members who are not engaged in care were not represented. Additionally, this analysis did not include the perspectives of people with lived experience related to HIV-related care, opioid use, or mental health or from primary opioid treatment providers, which will be crucial for the next phase of this research.

## Conclusions

In this study, HIV care providers identified a high burden of comorbid substance use disorders and mental health challenges, but experienced multiple barriers in connecting patients to care for these challenges. These included syndemic social challenges faced by patients such as housing insecurity and difficulties with transportation, lack of appropriate referrals, and stigma within the health system. Few clinics offered integrated, in-house treatment options for OUD or mental health, often due to high burdens placed on current providers and lack of resources for new services. Future efforts must emphasize identifying strategies to overcome these barriers and implement evidence-based strategies for HIV, OUD, and mental health treatment integration.

## Author Contributions

**Conceptualization:** Brandon A. Knettel, Hillary Chen, Elena Wilson, Mehri S. McKellar, Susan Reif.

**Formal analysis:** Brandon A. Knettel, Hillary Chen, David Agor.

**Funding acquisition:** Brandon A. Knettel, Elena Wilson.

**Investigation:** Brandon A. Knettel, Hillary Chen, Elena Wilson, Susan Reif.

**Methodology:** Brandon A. Knettel, Hillary Chen, Elena Wilson, Mehri S. McKellar, Susan Reif.

**Project administration:** Brandon A. Knettel, Hillary Chen, Elena Wilson, Mehri S. McKellar.

**Supervision:** Mehri S. McKellar, Susan Reif.

**Visualization:** Brandon A. Knettel.

**Writing – original draft:** Brandon A. Knettel, Hillary Chen, Elena Wilson, David Agor, Mehri S. McKellar, Susan Reif.

**Writing – review & editing:** Brandon A. Knettel, Hillary Chen, Elena Wilson, David Agor, Mehri S. McKellar, Susan Reif.

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
