## [Decision Letter · Decision Letter 0]

9 May 2024

PONE-D-24-07882Exploring strategies for integrating screening and treatment for opioid use disorder and mental health in HIV Care: A qualitative study with providersPLOS ONE

Dear Dr. Knettel,

Thank you for submitting your manuscript to PLOS ONE. After careful consideration, we feel that it has merit but does not fully meet PLOS ONE’s publication criteria as it currently stands. Therefore, we invite you to submit a revised version of the manuscript that addresses the points raised during the review process.

We look forward to receiving your revised manuscript.

Kind regards,

Sairah Hafeez Kamran, PhD

Academic Editor

PLOS ONE

2. In this instance it seems there may be acceptable restrictions in place that prevent the public sharing of your minimal data. However, in line with our goal of ensuring long-term data availability to all interested researchers, PLOS’ Data Policy states that authors cannot be the sole named individuals responsible for ensuring data access (http://journals.plos.org/plosone/s/data-availability#loc-acceptable-data-sharing-methods).

Reviewers' comments:

Reviewer's Responses to Questions

**Comments to the Author**

1. Is the manuscript technically sound, and do the data support the conclusions?

Reviewer #1: Yes

Reviewer #2: Yes

Reviewer #3: Yes

2. Has the statistical analysis been performed appropriately and rigorously? 

Reviewer #1: N/A

Reviewer #2: N/A

Reviewer #3: N/A

3. Have the authors made all data underlying the findings in their manuscript fully available?

Reviewer #1: Yes

Reviewer #2: Yes

Reviewer #3: Yes

4. Is the manuscript presented in an intelligible fashion and written in standard English?

Reviewer #1: Yes

Reviewer #2: Yes

Reviewer #3: Yes

5. Review Comments to the Author

Reviewer #1: This is an interesting study. The paper is well written and structured. However, the results do not clearly answer the aims of the study as well as are not quite related to the topic. The results showed that (1) the current lack of OUD and mental health screening and treatment in HIV care facilities, (2) situation of substance use among HIV patient is high, but opioid use was emphasized less often, (3) barriers to improve HIV patient care were, patient with mental health issue(s), financial and social challenges, no stable housing, transportation, stigma of mental health and substance use, limitation of referral system; which look like general barrier for HIV care (with or without any comorbidity). As mentioned before, the results do not clearly show barriers and strategies for the implementation of screening and treatment of OUD and metal health issues. Moreover, in the discussion part, some details seem over-mentioned from those reported in the results section. For example, strong consensus, rapid increase of OUD.

There are some more comments that are offered with the intention of helping to strengthen the presentation of the study.

Abstract:

- The authors said that in-depth interviews were conducted with 21 HIV treatment providers and social services providers; however, in the METHODS section showed individual semi-structure interview and group interview. The authors should provide more specific information about the method used in the abstract.

- May add sample recruitment method and study setting in the abstract.

- The results did not clearly answer the aims to the study.

Introduction

- May add more information/references regarding strategies and barriers of integrating screening and treatment of OUD and mental health in HIV patients.

Methods:

- The authors reported that purposive samples were recruited, please provide more specific characteristics of sample that will be used to recruit samples.

- The interview guides did not really show major questions that were used on the interview, only prompts are shown. Moreover, there were around 30 prompt questions in the table, is 30-60 minutes enough to obtain details and rich understanding of the topic?

Results:

- Organizational characteristics: the theme seem to present characteristics of HIV care facilities, however, the data only showed provided services of the facilities. The authors may present more characteristics of the facilities (such as number of health care providers/specialists, number of staffs) and relation of organizational characteristics and provided services (if possible)

Discussion

- 3rd Paragraph, the authors said that there was a rapidly increasing burden of OUD in HIV patients, but in the result part stated that “opioids were emphasized less often”

- Please discuss the limitation of the study regarding conducted both individual and group interviews.

Reviewer #2: The manuscript is well written and the few corrections identified are given as comments in the attached manuscript.

However, the reference section needs to be verified with reference to the journal referencing guidelines for maximum authors. and reference style.

Reviewer #3: The authors aimed to explore the potential benefits, barriers, and strategies for implementing opioid screening and treatment, along with mental health treatment, in HIV care. The methodology employed is transparent and valid for the stated objective. The findings of the manuscript are of interest to the readers of the Journal. However, I have the following minor concerns regarding the study results and title:

Comment:

1. The title of the study is "Exploring strategies for integrating screening and treatment for opioid use disorder and mental health in HIV Care: A qualitative study with providers." However, the study results do not primarily focus on strategies; instead, they emphasize identifying barriers. Therefore, it is recommended to revise the title to "Identification of barriers for integrating screening and treatment for opioid use disorder and mental health in HIV Care: A qualitative study with providers." Similarly, the objectives should be adjusted to remove references to strategies. Alternatively, the theme of strategies could be addressed in the results section and discussed in detail.

2. In Table 2, the age unit (Age in years) should be mentioned.

6. PLOS authors have the option to publish the peer review history of their article (what does this mean?). If published, this will include your full peer review and any attached files.

Reviewer #1: No

Reviewer #2: **Yes: **Abdul Nazer Ali

Reviewer #3: No

---

## [Author Response · Author response to Decision Letter 0]

13 May 2024

Please see attached Response to Reviewers document.

---

## [Decision Letter · Decision Letter 1]

22 May 2024

PONE-D-24-07882R1Provider perspectives on screening and treatment for opioid use disorder and mental health in HIV Care: A qualitative studyPLOS ONE

Dear Dr. Knettel,

Thank you for submitting your manuscript to PLOS ONE. After careful consideration, we feel that it has merit but does not fully meet PLOS ONE’s publication criteria as it currently stands. Therefore, we invite you to submit a revised version of the manuscript that addresses the points raised during the review process.

We look forward to receiving your revised manuscript.

Kind regards,

Sairah Hafeez Kamran, PhD

Academic Editor

PLOS ONE

Journal Requirements:

Reviewers' comments:

Reviewer's Responses to Questions

**Comments to the Author**

1. If the authors have adequately addressed your comments raised in a previous round of review and you feel that this manuscript is now acceptable for publication, you may indicate that here to bypass the “Comments to the Author” section, enter your conflict of interest statement in the “Confidential to Editor” section, and submit your "Accept" recommendation.

Reviewer #1: All comments have been addressed

Reviewer #2: All comments have been addressed

Reviewer #3: All comments have been addressed

2. Is the manuscript technically sound, and do the data support the conclusions?

Reviewer #1: (No Response)

Reviewer #2: Yes

Reviewer #3: Yes

3. Has the statistical analysis been performed appropriately and rigorously? 

Reviewer #1: (No Response)

Reviewer #2: N/A

Reviewer #3: Yes

4. Have the authors made all data underlying the findings in their manuscript fully available?

Reviewer #1: (No Response)

Reviewer #2: Yes

Reviewer #3: No

5. Is the manuscript presented in an intelligible fashion and written in standard English?

Reviewer #1: (No Response)

Reviewer #2: Yes

Reviewer #3: Yes

6. Review Comments to the Author

Reviewer #1: (No Response)

Reviewer #2: (No Response)

Reviewer #3: The authors of the manuscript has satisfactorily revised the manuscript as per the reviewers comments.

7. PLOS authors have the option to publish the peer review history of their article (what does this mean?). If published, this will include your full peer review and any attached files.

Reviewer #1: No

Reviewer #2: No

Reviewer #3: No

---

## [Author Response · Author response to Decision Letter 1]

23 May 2024

See attached response to reviewer comments document.

The minor revisions requested can be found in the file "Opioid HIV Manuscript_R2_tracked" on pages 6 and 7. The revisions have been marked using Track Changes.

---

## [Editor Report · Decision Letter 2]

27 May 2024

Provider perspectives on screening and treatment for opioid use disorder and mental health in HIV Care: A qualitative study

PONE-D-24-07882R2

Dear Dr. Knettel,

We’re pleased to inform you that your manuscript has been judged scientifically suitable for publication and will be formally accepted for publication once it meets all outstanding technical requirements.

Kind regards,

Sairah Hafeez Kamran, PhD

Academic Editor

PLOS ONE

---

## [Editor Report · Acceptance letter]

13 Jun 2024

PONE-D-24-07882R2 

PLOS ONE

Dear Dr. Knettel, 

I'm pleased to inform you that your manuscript has been deemed suitable for publication in PLOS ONE. Congratulations! Your manuscript is now being handed over to our production team.

Kind regards, 

on behalf of

Dr. Sairah Hafeez Kamran 

Academic Editor

PLOS ONE